# Immunological Perspective for Ebola Virus Infection and Various Treatment Measures Taken to Fight the Disease

**DOI:** 10.3390/pathogens9100850

**Published:** 2020-10-17

**Authors:** Sahil Jain, Svetlana F. Khaiboullina, Manoj Baranwal

**Affiliations:** 1Department of Biotechnology, Thapar Institute of Engineering & Technology, Patiala 147004, Punjab, India; drsahiljain88@gmail.com; 2Department of Microbiology and Immunology, University of Nevada, Reno, NV 89557, USA; 3Institute of Fundamental Medicine and Biology, Kazan Federal University, 420008 Kazan, Tatarstan, Russia

**Keywords:** T-cell immunity, bystander apoptosis, cytokines response, host immune evasion, Ebola vaccines

## Abstract

Ebolaviruses, discovered in 1976, belongs to the *Filoviridae* family, which also includes Marburg and Lloviu viruses. They are negative-stranded RNA viruses with six known species identified to date. Ebola virus (EBOV) is a member of *Zaire ebolavirus* species and can cause the Ebola virus disease (EVD), an emerging zoonotic disease that results in homeostatic imbalance and multi-organ failure. There are three EBOV outbreaks documented in the last six years resulting in significant morbidity (>32,000 cases) and mortality (>13,500 deaths). The potential factors contributing to the high infectivity of this virus include multiple entry mechanisms, susceptibility of the host cells, employment of multiple immune evasion mechanisms and rapid person-to-person transmission. EBOV infection leads to cytokine storm, disseminated intravascular coagulation, host T cell apoptosis as well as cell mediated and humoral immune response. In this review, a concise recap of cell types targeted by EBOV and EVD symptoms followed by detailed run-through of host innate and adaptive immune responses, virus-driven regulation and their combined effects contributing to the disease pathogenesis has been presented. At last, the vaccine and drug development initiatives as well as challenges related to the management of infection have been discussed.

## 1. Introduction

The *Filoviridae* family includes enveloped, non-segmented, negative–strand RNA ebolaviruses belonging to the genus *Ebolavirus* [1]. Until recently, six *Ebolavirus* species have been identified out of which, members of four species are known to be human pathogens [2]. These viruses are Ebola virus (EBOV), Sudan virus (SUDV), Taï Forest virus (TAFV) and Bundibugyo virus (BDBV) belonging to *Zaire ebolavirus*, *Sudan ebolavirus*, *Taï Forest ebolavirus* (earlier known as *Ivory coast ebolavirus* or *Côte d’Ivoire ebolavirus*) and *Bundibugyo ebolavirus* species, respectively [3]. EBOV is the most pathogenic and responsible for “The Great Outbreak of West Africa” in 2014–2016, the largest known ebolavirus outbreak [2,4]. Reston virus (RESTV) and Bombali virus (BOMV), members of *Reston ebolavirus* and *Bombali ebolavirus* species respectively, are not reported to infect humans [5,6,7].

Ebola virus disease (EVD), earlier known as Ebola hemorrhagic fever (EHF), is a fatal illness that has been described in humans as well as primates such as monkeys, chimpanzees and gorillas (reviewed in [8]). EVD is zoonotic wherein humans become infected when contacting infected animals or handling bushmeat [9]. Human-to-human transmission was reported as well via contact with blood and body fluids from infected individuals [10]. The disease severity as well as devastating social and economic effects led to EBOV classification as a Category A Priority pathogen by National Institute of Allergy and Infectious Diseases (NIAID), and as a “select agent” by US federal regulations and a bioterrorism Category A Agent by the Centers for Disease Control and Prevention (CDC) [11,12,13].

EBOV infection leads to lymphopenia, neutrophilia, increased serum proinflammatory cytokines, disseminated intravascular coagulation (DIC), liver necrosis and lymphoid tissue necrosis (reviewed in [8,14]). Additionally, innate and adaptive (humoral and cellular) immune responses were shown to be activated during the infection (reviewed in [15]). The infection results in the activation of type-I effector mechanisms (cytotoxicity and interferon production) as well as strong T cell activation and differentiation (reviewed in [16]). Similar results were obtained using non-human primates (NHP) model, the ‘gold-standard’ animal model of EBOV infection, (reviewed in [17,18]) confirming the role of the immune response in pathogenesis of the disease.

Management of EBOV infection remains a great challenge, where lack of a specific treatment is the main cause of high mortality. Therefore, vaccines remain the best approach to control the outbreaks and prevent death. Following 2014 West African epidemic, Russia and China had regionally licensed two Ebola vaccines [19] to counter the possible regional outbreaks. However, the efficacy of these vaccines was under debate as sufficient phase III trial data is not available [20]. During the 2018 outbreak in Democratic Republic of Congo (DRC), the Ervebo vaccine (Ebola Zaire Vaccine, Live), which is based on recombinant vesicular stomatitis virus (rVSV) vector, was recommended for compassionate and investigational use in the areas with the highest risk of EVD [21]. This vaccine conferred immunogenicity against EBOV during the 2014 epidemic, prompting the European Medicine Agency (EMA) to pass a ‘conditional marketing authorization’ for the Ervebo vaccine on 18th October, 2019 [21]. Ervebo was approved by the Food and Drug Administration (FDA) on 19th December, 2019 as the first licensed vaccine against EBOV [22]. Currently, despite the encouraging news on the vaccine approval, there is a long way to complete, global protection against future EBOV outbreaks.

## 2. Ebola Virus Disease (EVD)

### 2.1. Cell Targets

EBOV can infect almost any cell, except lymphocytes (reviewed in [23]); however, virus propagation was demonstrated in limited cell types such as fibroblasts, endothelial cells, NK cells, epithelial cells and hepatocytes (reviewed in [24]). Within these cell range, it appears that mononuclear phagocytes (macrophages including alveolar cells, monocytes and dendritic cells) are the most susceptible to infection in the initial phase of disease [25,26,27]. Dendritic cells (DC) expressing DC-specific intercellular adhesion molecule (ICAM)-3-grabbing nonintegrin (DC-SIGN) were shown to be susceptible to infection as well, whereas langerin expressing cells (CD141^+^ DC of mucosal epithelium and skin Langerhans cells), epidermal DCs and mucosal epithelium DCs were found to be resistant (reviewed in [28]). Mononuclear phagocytes moving out of lymph nodes and spleen are thought to be responsible for viral dissemination [26].

### 2.2. EBOV Attachment and Entry

Three uptake mechanisms were identified by which EBOV could enter the cell: macro-pinocytosis, lipid raft and receptor-mediated endocytosis [23]. Reports suggest that class I phosphastidylinositol-3-kinase Akt pathway [29] and cholesterol enriched lipid raft microdomains [30] are essential for viral trafficking across the cell membrane. Additionally, multiple receptors were identified as contributing to virus attachment including glycosaminoglycans, folate receptor α, β1 integrin receptor, human macrophage galactose-and N-acetylgalactosamine-specific C type lectin (hMGL), DC-SIGN, triggering receptor expressed in myeloid cells 1 (TREM-1) and various other C-type lectins [18,31,32,33,34,35]. EBOV is also known to interact with T-cell immunoglobin mucin domain (TIM) and Tyro3/Axl/Mer (TAM) receptors [17]. None of these receptors has been identified as critical for viral attachment.

Endocytosis precedes the uncoating and fusion between viral and endosomal membranes (reviewed in [36]). Internalization of the virus into a macropinosome is followed by its transfer to an endosomal compartment containing the cysteine proteases such as cathepsin B and cathepsin L [37]. The latter helps to digest the viral glycoprotein (GP) which initiates viral and endosomal membrane fusion [37]. Processed GP1 interacts with Neimann-Pick C1 (NPC1) protein of the late endosome (an extremely crucial interaction for EVD) which results in coupling of the virus envelope and membrane of the endosome [38]. The fusion loop domain of GP2 helps to initiate the membrane fusion by inserting into the host endosomal membranes [39]. GTPases, especially Rab7 GTPase (related to the late endosomes), play a crucial role in viral fusion [40]. A low pH dependent endosomal function is required by cathepsin digested GP1 subunit for fusion [41]. Therefore, fusion process can be blocked by the cysteine protease determents and increasing the pH of the environment [41]. After fusion, trimeric heptad regions of the GP2 subunit form a hexa-helical transmembrane structure, facilitating the release of viral proteins into the host cytoplasm where they initiate the replication process [42].

### 2.3. EVD Symptoms

EVD symptoms appear after an incubation period ranging from 2–21 days [14]. They include fever, weakness, vomiting, anorexia, abdominal pains, chills, “ghost-like” expressionless face, proteinuria and diarrhea (reviewed in [12,43]). Infection results in viremia which presents with hemorrhages (petechiae, mucosal hemorrhage, ecchymosis and visceral hemorrhage), tachycardia, electrolyte disorders, multiple vital organ (liver, respiratory and renal) failure and necrosis (reviewed in [23,44,45]). Multi-organ failure is the primary cause for death in EVD [23] while electrolyte imbalances such as hypocalcemia, hyponatremia and hypomagnesemia may lead to cardiac arrhythmias or coma [28,46,47].

Surprisingly, hemorrhage has been recorded in only 20% cases since the first outbreak and may be, partially, due to the hepatocellular necrosis, which impairs production of functional blood coagulation enzymes in liver [28,48,49]. Additionally, genetic and nutritional factors were suggested to contribute to hemorrhagic syndrome [28]. Interestingly, Mc Elroy et al., reported lack of correlation between hemorrhage and fatality [50]. In addition to hemorrhages, virus was shown to cause primary pulmonary infection [51], neurological complications such as seizures and tremors [52], ocular disorders such as conjunctivitis and uveitis [53] and affect the liver function [54]. Rheumatological manifestations such as arthritis, arthralgia, tendinitis, bursitis and myalgia are common in patients [55]. It appears that virus could persist in the immune privileged organs long after the clearance [17,56]. This assumption was supported by the detection of viral RNA in the urine, sweat and other body parts indicating the viral persistence long after the clinical symptoms subside [56].

## 3. Host Immune Response

A comprehensive diagram of the EBOV effects on the host immune response is presented in Figure 1.

### 3.1. Mononuclear Phagocytes

EBOV can affect the mononuclear phagocytes differentiation, which could interfere with their ability to recognize and present the antigen [26]. Additionally, EBOV infection of mononuclear phagocytes causes the upregulation of anti-apoptotic genes such as neuronal apoptosis inhibitory protein (NAIP) and cellular inhibitor of apoptosis protein 2 (cIAP2) [26]. Additionally, EBOV infection inhibits DC maturation which could affect the activation of an innate and adaptive immunity [18]. Interestingly, an in vitro analysis of monocyte-derived DC revealed that these cells can survive for days after infection [57], while supporting virus replication. Therefore, it was suggested that DC could serve as a vehicle disseminating the virus.

### 3.2. Cell Mediated Immune Response

#### 3.2.1. T-Cell Response

Multiparametric flow cytometry analysis has revealed a robust activation of T_c_ cells followed by a substantial proliferation in fatal as well as survivor cases [28,56,58]. An average of 45% CD8^+^ T cells, consisting of HLA-DR^+^/CD38^+^, Ki-67^+^/granzyme B^+^ and Ki-67^+^/PD-1^+^ subsets, were found to express activation markers HLA-DR, Ki-67 and CD38 [56]. NHP studies focused on EBOV GP presented contrasting results regarding the indispensability of CD8^+^ T cells for EBOV infection survival [59,60]. In 2018, Sakabe and coworkers demonstrated that memory CD8 T cells secrete interferon gamma (IFN-γ) and tumor necrosis factor alpha (TNF-α) in nearly 80% survivor subjects, especially upon activation with proteins other than GP viz., EBOV nucleoprotein, virion protein 24 (VP24) or VP40 [61]. Analysis of samples collected during the West African epidemic (2014–2016) by Speranza and co-workers revealed the abundance of T-cell immunity transcripts (RANTES, CD40L, CD28 etc.) in survivors and of T cell homeostasis drivers (PD-1 and Indoleamine 2,3-dioxygenase) in fatal cases, thus, confirming the notion of a robust and sustained T-cell response mounted in survivors [62]. Moreover, survivors show a characteristic chemokine (C-C motif) ligand 5 (CCL5/RANTES) expression, further, supporting the role of T cells in viral clearance. In contrast, the fatal cases are devoid of T cell viral clearance and present a clustering of T cells in gut and respiratory mucosa [62].

Immunophenotyping analysis of fatal and survivor blood samples indicated activation of CD8^+^ and CD4^+^ T cells. Still, the magnitude and diversity of the immune response induced in the survivors were more robust as compared to fatal cases. Proliferation of T cells and yet their failure to effectively protect against EVD in fatal cases may be attributed to either a state of T-cell exhaustion [58,63,64] or little and delayed proliferation in some cases owing to exaggerated viral count and uncontrolled viral replication [28,56]. Additionally, activation of Ki-67^+^/PD-1^+^ CD8^+^ T cell subset seems responsible for a weaker adaptive immune response via PD-1 inhibitory pathway [56]. Indeed, oligoclonal response and greater expression of CTLA-4 and PD-1 in CD8^+^ and CD4^+^ T cells was found in fatal cases, which could explain the high virus titer as well as T-cell exhaustion [58,62,63,65,66]. Whether the high viral titers and inflammation could cause a greater CTLA-4 and PD-1 expression on T cells in fatal cases remains unclear [58].

#### 3.2.2. Lymphocyte Apoptosis

A massive reduction of lymphocyte (CD8^+^, CD4^+^ and NK cells) counts was found in the initial as well as the end stages of fatal EVD [67]. Experimentally infected NHP models displayed a reduction in peripheral NK cell count which may be attributed to apoptosis [68]. CD8^+^ and CD4^+^ cells appear to be the most affected as their counts in fatal cases were found reduced to an approximately one fourth their number in survivors [23]. Expression analysis of CD95 for CD8^+^ and CD4^+^ T cells as well as PD-1 for CD4^+^ T cells in EVD patients suggested the role of apoptotic pathways in massive lymphocyte loss [69]. Lymphocyte apoptosis could be attributed to (a) deregulation of DC/T interaction, i.e., lack of co-stimulatory rescue signals by malfunctioning DCs or (b) upregulation of apoptotic genes such as Fas, Fas Ligand (FasL) and tumor necrosis factor (TNF)-related apoptosis-inducing ligand (TRAIL) in infected leukocytes or (c) Direct lysis by EBOV GP [23,67,70]. Indeed, excess TNF-α secretion is thought to contribute to lymphocyte apoptosis in NHP models [71].

However, bystander T-cell apoptosis doesn’t seem to be a definite EVD characteristic and as such, does not seem to indicate fatality. This is supported by a report that EBOV caused death in apoptotic gene knockout mice despite reduced T cell apoptosis [72]. Additionally, T cells produced in experimentally EBOV infected mice were found to protect naive mice upon adoptive transfer [73] while complete depletion of T cells in experimentally infected NHP led to an increased fatality rate [59] suggesting a significant role of T cells in host survival. Similar data was demonstrated using mice deficient in cytotoxic T cells (T_c_ cells), where mice were dying upon experimental EBOV infection [74].

Overall, various reports suggest a critical, though highly varied role of T cells/cell mediated immune response upon EBOV infection in both, fatal and survivor cases and therefore, the actual lymphocyte apoptosis mechanism, T cell immunity dynamics and behavior during EVD is vital to understanding EBOV pathology.

### 3.3. The Cytokine Storm

Multiple studies indicate a surge in procoagulant tissue factor protein, oxygen free radicals, cytokines and chemokines upon EBOV infection which could lead to a severe disease and death [75,76,77,78,79]. Exact mechanisms behind the early cytokine storm are unclear as in vitro analysis of infected human DC and macrophages indicated the restriction in their ability to secrete the immune mediators [80]. Elevated levels of several cytokines were found in macrophages infected with EBOV even before detection of viral gene expression, indicating a role of EBOV GP in inducing the inflammation [81]. Another in vitro study has reported that shed GP is capable of binding to uninfected mononuclear phagocytes via TLR4 receptors and contributing to the cytokine production [82].

Intracellular cytokine assay analysis revealed the production of IFN-γ by CD8^+^ T cells in EBOV infected mice [73]. Additionally, in another study using a mouse model, EBOV was shown to infect CD4^+^ T cells without replication, while inducing cytokine release [79]. A “superagonist-like” effect regarding NFAT signaling pathway activation was demonstrated upon binding of EBOV to CD4^+^ T cells, enabling these leukocytes to secrete IFN-γ, IL-2, TNF-α and IL-8 [79]. Similarly, analysis of human peripheral blood mononuclear cells (PBMCs) obtained from convalescent EBOV patients indicated high IFN-γ secretion by CD8^+^ T cells [56]. Though CD4^+^ T cells were found less capable of secreting IFN-γ than CD8^+^ T cells, still, the majority of such CD4^+^ T cells were also secreting IL-2 and TNF-α [56]. Therefore, a combined effect of virus affected macrophages and activated T cells along with uncontrolled viral replication has been contemplated as a plausible reason for massive pro-inflammatory cytokine production [79].

Increased levels of IL-1β, IL-6, IL-10, IL-15, IL-16, TNF-α, IFN-α, IFN-β, IFN-γ, NO, MIP-1α, MIP-1β, MIF and IP-10 are primarily demonstrated in EVD cases [83,84]. Increased IL-10 and TNF-α levels are universally accepted as EVD fatality indicators whereas association of increased NO, IFN-α, IFN-γ, IP-10, IL-12, IL-17 and IL-6 levels with fatality has mixed recognition [18,23,85,86,87]. An early IFN-γ response followed by an extensive lymphocyte apoptosis is an accepted fatality indicator [23,83], whereas high IFN-β serum levels could indicate lower disease severity [17]. Higher expression of CTLA-4 in T cells corresponded to presence of greater TNF-α and IL-8 quantities in fatal cases. Additionally, serum levels of certain chemokines, such as MIP-1α, MIP-1β and MCP1, were substantially higher in fatal EVD cases [58]. It appears that an increased levels of MIP-1α and IP-10 are associated with hemorrhages, which could lead to a serious complications and death in EVD (reviewed in [23]). Though early and short-lived IFN-α production has been related to survival [88], it appears that the virus can counteract it (discussed in Section 4).

### 3.4. Interplay between Cytokines and Coagulation Factors

As discussed, severe infection leads to hyperproduction of proinflammatory cytokines. These cytokines activate coagulation factors such as thrombomodulin, ferritin etc. [23]. A study reported upregulation of procoagulant protein tissue factor in endothelial cells and monocytes by TNF-α and IL-6 [89]. The released coagulation factors in turn upregulate proinflammatory cytokines. Studies have suggested that fibrin fragment E and thrombin induce IL-6 production in monocytes while thrombin induces IL-6 and IL-8 production in endothelial cells [18,90]. Hence, a deadly chain reaction ensues upon filoviral infection which culminates into shock, vascular damage and homeostatic imbalance.

### 3.5. Endothelial Cell Dysfunction and Vascular Damage

Endothelial cells seem to be directly infected during terminal stages of EBOV infection due to over expression of proinflammatory cytokines but do not seem to exhibit any structural damage [18,28]. EBOV GP, supported by TNF-α, is thought to play a pivotal role in endothelial cell dysfunction, consequently, leading to anoikis and hemorrhage [23,91,92,93]. A study during 1995 outbreak found antigens in endothelial cells in different body tissues [94]. A recent study supported endothelial dysfunction on basis of increased thrombomodulin, P-selectin and PE-CAM (all are markers of endothelial activation and dysfunction) in patients [28,50]. However, the precise timing and consequences of endothelial cell dysfunction are yet to be elucidated. An early study reported no antigen presence in endothelial cells [95]. Additionally, antigen multiplication in vascular endothelial cells in later stages (after appearance of hemorrhage) upon experimental EBOV infection of cynomolgus monkeys has been reported [18,26].

Disseminated intravascular coagulation (DIC) coupled with low platelet count as well as coagulation factor deficiency is known to occur during EBOV infection [96]. It has been debatably related to endothelial cell disorders, especially release of thrombomodulin by activated endothelial cells [18,28,84]. Various reports attribute the endothelial cell activation and consequent dysfunction as well as vascular damage to (a) release of proinflammatory cytokines [75,77,97], especially TNF-α [98,99] and NO [87] or (b) overexpression of cell surface tissue factor in monocytes and macrophages [26] or (c) elevated levels of Von Willebrand factor (vWF), a protein which acts as mediator between platelets and endothelial cells [28].

### 3.6. Humoral Immune Response

Although cell-mediated immunity is suggested to play the leading role in protection against EBOV infection [17], humoral immunity, such as production of antibodies coupled with T_c_ activation (via Fas/FasL or perforin pathway) is suggested to be a credible marker of survival [23,56]. EBOV-specific IgM and IgG antibodies are detected nearly 12–31 days and 23–42 days post infection, respectively [56,86]. Humoral response is thought to play a lesser role in EVD survival as susceptibility to infection and recovery in mouse models was independent of humoral response [74]. Additionally, no difference in antibody titers was found in survivors and fatal cases during the outbreak [100]. Little role of neutralizing antibodies (N_Ab_) in recovering from acute stages, attributed to the need for high diversity in N_Ab_ repertoire and to a lack of early B cell affinity maturation, has been reported during the course of infection [28,56,101,102]. In another report, some fatal cases have been recorded despite the presence of antibodies while several survivors did not develop IgG [103].

## 4. Evasion of Host Immune Response by EBOV

Multiple studies have demonstrated that EBOV employs an array of mechanisms to effectively evade the host immune responses (Figure 2), where viral GP, VP24, VP35 and VP40 play a significant role (reviewed in [104]). The efficacy of host immune evasion by EBOV is indicated by the rapid replication rate displayed by the virus inside the host cells [27]. VP35 and VP24 can interfere with the host antiviral defense system by restricting early IFN production and signaling (reviewed in [105]). VP35 and VP24 regions responsible for virus protection against host antiviral activity are called innate response antagonist domains (IRADs) [105].

VP24 can bind to specific members of karyopherin alpha (KPNA) family proteins involved in nuclear translocation of phosphorylated signal transducer and activator of transcription 1 (STAT 1) [106]. This could be explained by the fact that the binding affinity of VP24 to KPNA is significantly higher than that of phosphorylated STAT 1 [106]. It renders karyopherin-α1 incapable of transporting phosphorylated STAT 1 across the nuclear pore to activate gene transcription. This results in interference with the type I and II IFN response by restricting the JAK-STAT to the cytoplasm [105,107,108] (Figure 2). However, EBOV, SUDV and TAFV inhibit IFN-signaling with greater efficacy as compared to BDBV and RESTV, a feature attributed to high KPNA binding ability of the former viruses [104,109]. A host cell-type dependent restriction of IFN signaling through mitogen-activated protein kinases (MAPK) pathway by VP24 has also been reported [105,110]. VP24 was shown to block p38 phosphorylation in human embryonic kidney 293 (HEK 293) cells but not in HeLa cells, resulting in blockage of MAPK pathway only in these cells [105,110] (Figure 2).

VP35 plays a central role in host immune response evasion (Figure 2). Inhibition of type I IFN production by VP35 is a pan-filovirus characteristic [104]. VP35 effectively shields dsRNA from detection during replication/transcription stage [105]. It caps dsRNA via Phe235 and Phe239 residues and thus, prevents host retinoic acid-inducible gene I (RIG-I) and melanoma differentiation-associated protein 5 (MDA-5) from identifying viral dsRNA [111,112]. VP35 can also block host protein activator of the interferon-induced protein kinase (PACT) protein activation in RNA-independent manner, in effect cutting off RIG-I or MDA-5 signaling and IFN-α/β gene expression [108,113]. Interestingly, the same VP35 carboxy-terminal domain (residues 220–340) is involved in capping the ends of dsRNA, VP35-dsRNA interaction (especially basic residues such as Arg312, Lys319 and Arg322) and VP35-PACT interaction (especially residues Arg312, Arg322 and Phe239) [112,113,114]. Due to the first two functions, this VP35 carboxy-terminal domain is also known as interferon inhibitory domain (IID) as it shields dsRNA from host cellular sensors [105,108]. This domain is involved in restriction of the RIG-I ATPase and IFN-β promoter activity [104]. Additionally, VP35 IID domain, especially Arg305, Lys309 and Arg312 residues prevent dsRNA-dependent protein kinase R (PKR) activation in dsRNA-binding independent manner. This blocks eukaryotic translation initiation factor 2α (eIF2α) phosphorylation, thus, enabling continuation of viral protein synthesis [104,115,116] (Figure 2).

VP35 can also block activation of IKKε and TBK1 kinases by interacting with their conserved domains [117]. This, in effect, blocks IRF-3 and IRF-7 phosphorylation by IKKε and TBK1 kinases. Prevention of IRF-3 phosphorylation could hamper IFN production, as this transcription factor is essential for induction of IFN-β promoter [117]. Additionally, VP35 has the ability to increase IRF-3/7 SUMOylation [118]. Both mechanisms prevent IRF-3 and IRF-7 mediating signaling and the subsequent IFNα/β gene expression [108,119,120].

The suppression of an innate immune response by VP35 has far reaching consequences. It impairs RIG-I-like receptor (RLR) signaling resulting in lack of DC maturation which restricts IFN-α/β and cytokine production as well as T cell activation [108,121]. This ultimately impedes T cell response and may lead to a collective failure of an adaptive immune response (reviewed in [108]). Interestingly, blocking DC activation requires IRAD of both, VP35 and VP24 (reviewed in [105]).

It appears that secreted glycoprotein (sGP) could also contribute to the immune response evasion by acting as a decoy for anti-EBOV antibodies. Mohan and coworkers demonstrated that sGP can function as a roadblock between the virus GP and anti-GP 1,2 antibodies as it can effectively compete for these antibodies [122]. This could prevent the interaction between antibodies and the virus GP and lead to antigenic subversion [122]. Additionally, GP1 glycosylation, although not required for the virus entry, could protect GP from immune recognition, thus, enabling binding to the host cell [123]. An interesting role of sGP in host leukocytes evasion was postulated by Wahl-Jensen et al., reporting that sGP decreased the permeability of infected vascular endothelial cells to TNF-α as well as restored their barrier function post exposure of the infected cells to viral GP [93].

EBOV GP can interfere with immune recognition of HLA class I and II molecules, a phenomenon termed as “steric occlusion” [124]. Not only does this occur on host cells, steric occlusion also prevents the recognition of EBOV GP [124]. Therefore, steric occlusion could be another way in which virus can elude the host immunity. EBOV GP also has the anti-tetherin activity, thus, disabling the stoppage of VP40-mediated viral budding by host cell tetherin as well as disabling the immune response stimulation via NF-kB signaling [125,126] (Figure 2).

## 5. EBOV: Prevention and Control

### 5.1. Vaccine Development against EBOV

The first vaccine candidates against EBOV were tested in the 1970s and 1980s which were based on immunization with an attenuated virus [127]. Lack of efficacy and concern for safety led to consideration of an alternative approach using DNA vaccines in the 1990s [128]. In recent years, multiple approaches were used to design an anti-EBOV vaccine which included various vectors such as alphavirus replicons [129,130,131], virus-like particles [132], human adenoviruses (Ad) [133,134,135], a biologically contained EBOV lacking VP30 [136], chimpanzee adenovirus [137], DNA [138], paramyxoviruses [139,140], cytomegalovirus (CMV) vectors [141], rabies virus [142,143], modified vaccinia virus Ankara (MVA) [144] and different strategies with recombinant vesicular stomatitis virus (rVSV) [145,146] (Table 1).

The rVSV vector based EBOV vaccine candidates are considered as having a high protective potential (Table 1). rVSV vector based vaccines can be administered via mucosal membrane [163] and are highly immunogenic [164,165] as proven in NHP models where a robust and protective cell mediated immune response was demonstrated when lethal dose of virus was used [145]. Another study reported complete protection of NHPs even when vaccine was used three months after Lassa virus VSV-based vaccine administration indicating there are no pre-existing immunity concerns to rVSV [166,167]. Indeed, a recombinant live-attenuated VSV vector based vaccine expressing EBOV GP was FDA approved on 19 December 2019 for individuals ≥ 18 years age [168,169]. Merck Sharp and Dohme Corp. received the authorization to market a single dose vaccine, under the tradename Ervebo [169]. Ervebo vaccine induces antibody response and CD8^+^ T-cell activation upon administration [168], though the efficacy of resultant antibodies in viral clearance needs further analysis. Before FDA approval of Ervebo vaccine, Farooq et al., reported a role of circular follicular T helper cells upon injection of rVSV-ZEBOV-GP vaccine candidate in human subjects [170]. A correlation between the frequency of circulating CXCR5^+^ CD4^+^ T-cells and antibody titers was also recorded [170].

Another approach for EBOV vaccine development includes a combination of MVA-BN Filo and rAd26-EBOV [171]. The combination was shown to be well tolerated, capable of activating CD8^+^ and CD4^+^ cell responses as well as generated viral neutralizing antibodies [171]. Additionally, this combination is currently in phase III clinical trials [160] (Table 1). The only vaccine candidate currently in phase IV clinical trials is a GamEvac-combi vaccine which consists of live-attenuated rVSV and Ad5-EBOVGP [161] (Table 1). This vaccine could activate CD8^+^ and CD4^+^ T cells which could explain its efficacy [162].

Another vaccine candidate is developed by removing the VP30 gene (EBOV*∆*VP30) from EBOV genome [172,173]. This recombinant virus is replication incompetent, yet it was shown to be safe and can induce a robust immune response in NHP [136]. Wild type as well as H_2_O_2_ treated EBOV*∆*VP30 vaccine conferred complete protection via producing high antibody titers directed against various viral proteins [136]. Production of IFN-γ producing mononuclear cells was also suggested though their significance and impact was not reported [136]. Another vaccine candidate, MVA based, replication-incompetent, modified to express EBOV GP and VP40 proteins has conferred 100% protection in guinea pig and macaque models against lethal EBOV infection and has currently entered clinical trials [144].

*Rabies virus* (RABV) infection has been observed in many regions endemic for EBOV infection [174]. Hence, it could be suggested that the recombinant RABV based vector vaccine could be used to protect against both, EBOV and RABV infection. RABV vector based vaccine was reported to induce complete protection by eliciting strong humoral immune response in NHP infected with EBOV when a single dose was used [142]. Modification of the inactivated RABV vaccine vector using a codon-optimized antigen [143] and use of adjuvants was also protective [175]. This strategy is moving forward to being tested in clinical trials [175].

Peptide based vaccines have emerged as alternative candidates which display improved safety, specificity, selectivity and stability. Antigenic peptides, containing immunogenic epitopes, represent the minimum requirement for recognition of pathogen by immune system and are capable of inducing a strong immune response (reviewed in [176]). In efforts to identify potential epitopes, computational prediction of H-2d-specific T cell epitopes and their identification in BALB/c mice infected with replication-incompetent, ebolavirus GP expressing adenovirus vectors was done for *Sudan ebolavirus* and *Zaire ebolavirus* [177]. This study identified RPHTPQFLF, GPCAGDFAF and LYDRLASTV as potentially immunogenic peptides in mice model [177]. In 2015, 28 ebolavirus 9-mer epitopes with no apparent human homologs were identified using computer based approach [178]. Additionally, TLASIGTAF peptide was identified as a highly potent B and T cell activating epitope with the ability to be a part of epitope-based vaccine as it showed positive interaction with 12 HLA alleles and population coverage of 80.99% [179]. Later, computational identification of highly conserved peptides containing T and B cell epitopes in GP as well as NP non-redundant sequences belonging to all reported strains of ebolaviruses infecting humans was done by our research group [180,181]. In silico selected NP peptides showed enhanced IFN-γ production in most of the blood samples in vitro [181].

To summarize, the vaccine approaches discussed so far present specific advantages. Non-replicating vectors such as Ad5 and EBOV*∆*VP30 are considerably safer than replication-competent vectors (reviewed in [182]). rVSV vector-based vaccines are highly immunogenic and seem to be devoid of pre-existing immunity concerns [145,167]. A combination of MVA-BN Filo and rAd26-EBOV seems capable of generating both, cell mediated and humoral immune response [171]. H_2_O_2_ treated EBOV*∆*VP30 vaccine candidate addresses the earlier safety concerns and has exhibited high experimental efficacy [182] while recombinant rabies virus-based vector offers an opportunity to fight against two diseases (rabies and EBOV) endemic to same geographical regions [174]. Peptide-based vaccines offer a safer, economical and faster approach to counter the rapidly increasing threat of endemic pathogens [176].

These vaccine development approaches also suffer from different limitations (reviewed in [183]). Production difficulties and high cost are the major limitations of using the VLP platform [182]. DNA vaccines require regular booster dosages (reviewed in [184]). Pre-existing immunity against adenovirus based vectors and consequently, low efficacy, presents a major concern [185]. Low immunogenicity and in vivo peptide instability are major challenges while designing peptide-based vaccines [176].

### 5.2. Post-Exposure Therapeutics Development against EBOV

The constant EBOV infection threat and highly contagious nature of the infection make the search for post-exposure therapeutics especially important. The need for EBOV therapeutics was further signified after the 2014 outbreak when WHO permitted the use of untested drugs to control the high death rate (reviewed in [186]). Currently, 80 drugs with anti-EBOV activity are consented by the FDA (reviewed in [187]). A brief summary of various works done with respect to therapeutics development is presented in Table 2.

One of the therapeutic approaches involves recombinant nematode anticoagulant protein c2 (rNAPc2) which helps combat the thrombin-related organ failure. In a study, only one third of the total EBOV exposed Rhesus macaques survived upon administration of rNAPc2, indicating only partial and limited protection elicited by rNAPc2 [188]. In Dec 2014, FDA granted orphan drug status to rNAPc2 and recommended it as post-exposure treatment [189] despite the limitations such as need for repeated dosages and low efficacy [190]. rNAPc2 has passed phase II clinical trials [190].

The therapeutic efficacy of the small RNA (sRNA) molecules was also tested using NHP model of EBOV infection. TKM-100802, a small RNA-interfering (siRNA) molecule targeting VP35 and L genes, was administered to NHP 30 min prior EBOV infection [191]. Protection was conferred in 100% animals which received a total of six doses during six consecutive days [191]. The same drug proved highly effective when given to EVD patients in Europe and US; however, no conclusive result could be declared as the patients received an additional therapeutics simultaneously [192]. TKM-130803 was formulated by modification of TKM-100802 to increase the drug efficacy towards the specific strain responsible for 2014–2016 EBOV outbreak [193]. However, this drug failed to increase the survival rates of EVD patients [193].

AVI-6002, a combination of 2 phosphorodiamidate morpholino antisense oligomers with piperazine residues along the oligomeric backbone, was designed to target EBOV VP35, VP24 and L proteins [194]. This drug showed 60% efficacy in NHP EBOV infection model [194] (Table 2). Additionally, it was well tolerated during phase I clinical trials [195,196]. Another drug was developed by BioCryst Pharmaceuticals, targeting L protein [197]. This drug is an adenosine analog named BCX4430 which is a non-obligate RNA-chain terminator [197]. BCX4430 has shown encouraging results in EBOV infected mice as well as NHP [197]. It also completed phase I clinical trials in 2016 (CT: NCT02319772). Dismal pharmacokinetic properties and short half-life of BCX4430 are the major drawbacks while on the upside, it did not become incorporated into the host DNA [196,197]. Another nucleoside drug capable of inhibition of viral RNA synthesis is Favipiravir (Table 2), which is produced by modification of pyrazine [198]. Favipiravir was shown effective in mice [199]; however, only decreased virus titer was reported when used to treat EBOV in NHP model [200]. Therefore, this drug is approved only for the compassionate use by the French drug safety agency [201].

Remdesivir or GS-5734 had 100% protection in NHP when administered 72 h post EBOV infection [202] (Table 2). This drug was used as an emergency drug in both, 2014-2016 epidemic and the 2018 DRC epidemic [203,204]. Remdesivir cleared phase II clinical trials in 2019 (CT: NCT02818582) and is currently a part of phase II/III multi-therapeutic investigation trial (CT: NCT03719586). However, despite experimental success, low efficacy in human trials has been observed [205].

Compound 7, a benzodiazepine derivative, was shown to bind directly to EBOV GP in cell culture, preventing EBOV entry [206]. Other compounds viz., NSC62914 (a reactive oxygen species) and LJ001 (a rhodanine derivative) have shown anti-EBOV activity in vitro and in mouse model [207,208]. Additionally, Tolcapone (a FDA approved drug) impaired EBOV replication in vitro while 3-deazaneplanocin A (a carbocyclic nucleoside) and FGI-103, FGI-104 and FGI-106 (broad-spectrum antiviral agents) were shown to protect EBOV infected mice [209,210,211,212].

Some compounds such as Benzylpiperazine adamantane diamides, Arbidol, wortmannin, rottlerin and latrunculin A were shown to act as virus entry inhibitors [196,204]. Clomiphene and toremifene were identified as EBOV entry inhibitors in vitro and protected 90% and 50% mice, respectively [204,213].

Other compounds with anti-EBOV efficacy include pradimicin A and benanomicin A, which are non-peptidic antibiotics that target ebolavirus GP (reviewed in [214]). Enzon Pharmaceuticals provides commercial-grade human mannose-binding lectin (MBL), a C-type lectin that identifies various ebolavirus surface glycan structures, such as glucose, mannose etc. (Table 2). Griffithsin (GRFT) is a red algae derived lectin which is devoid of mitogenic activity and interacts with the terminal mannose molecules of N-linked glycans on the outside of ebolavirus [215]. In contrast to many lectins, GRFT has limited effect on proinflammatory cytokine production in PBMC, has low toxicity and can be produced on a large scale [215] (Table 2).

Antibodies are known to inhibit viral entry as well as facilitate the antibody-directed cell-mediated cytotoxicity by NK cells (reviewed in [229]). Additionally, together with the complement system, antibodies can neutralize the virus [229]. Therefore, EVD convalescent plasma (CP) containing polyclonal EBOV specific IgG have a potential to inhibit virus replication and dissemination [230]. Additionally, this approach presents a feasible and safe way for EVD treatment during an outbreak [230]. CP could also help to manage the blood loss commonly found in EVD [231]. CP administration was used alongside other treatments the UK researcher, who had contracted the disease while working with samples collected during the 1976 outbreak [232] as well as for 8 cases during Kitwit outbreak in 1993 [231]. In the latter case, seven patients survived but conclusive evidence of CP efficacy is lacking as other treatments such as antibiotics, antipyretics, oral rehydration and chloroquine were also used [232]. During 2014–2016 outbreak, a phase II/III trial (CT: NCT02342171) was conducted to determine the efficacy of CP in Guinea where 84 EVD patients received up to 500 mL CP along with the standard of care (SOC) treatment [230]. Although, clinically significant difference was not found between these patients and 418 control group (EBOV patients receiving SOC only), the survival rate was found higher when CP was the part the patient management protocol [230]. Limitations of CP include pre-screening requirement for blood transmitted pathogens, transfusion reactions, CP harvest timing, toxicity related problems and lot-to-lot variations [47,232,233], which directed researchers towards developing monoclonal antibodies.

Monoclonal antibodies (mAb) are IgG antibodies produced by a single B cell clone that could help to control virus spread by targeting EBOV GP (reviewed in [190,234]). A human convalescent monoclonal antibody, KZ52, was shown to be effective in mouse models; however, it failed in protecting NHPs from EBOV infection even when injected one day prior to exposure [235]. Another mAb, MAb114, is human mAb isolated during the Kitwit outbreak in 1993 [236]. Though it conferred protection in NHP when administered 120 h post lethal EBOV infection, viremia and clinical signs were still present [234]. These symptoms were shown to be reduced by administering a cocktail of mAb114 and mAb100, another mAb isolated from same survivor [234] (Table 2). mAb114 had cleared phase I clinical trials (CT: NCT03478891) in 2019 and is currently in phase II/III clinical trials (CT: NCT03719586).

MB-003 is a cocktail of mAb developed by the US Army Medical Research Institute of Infectious Diseases and consists of three murine mAbs (13C6, 6D8 and 13F6) against EBOV GP [232]. After encouraging results using mouse model [237], the therapeutic efficacy of this cocktail was tested in NHP, where two third of animals survived the EBOV infection [238]. Additionally, the therapeutic efficacy of MB-003 was confirmed when used five days post infection, protecting 40% of NHP [225] (Table 2).

ZMAb, developed by the Public Health Agency of Canada, includes three murine mAbs (1H3, 2G4 and 4G7) [239]. In NHP model, ZMAb conferred 100% protection when animals received three doses of the drug in six days starting 24 h after infection [226] (Table 2). To check if any immune response was developed in NHP treated and recovered, they were infected again 2.5 months post first infection. All the macaques survived the second EBOV challenge, suggesting the presence of an immune protection for months after the ZMab treatment [240]. Combined treatment with ZMAb cocktail administered 96 h post infection followed by AD5-IFNα injected 24 h post infection was also effective in 50% NHP [241]. Further, the same study reported 100% protection of NHP when AD5-IFNα was administered single time together with the first ZMAb injected 72 h post infection [241]. Compassionate use of ZMAb along with other treatments such as CP and favipiravir has resulted in 100% survival in 6 EVD patients [234]. Interestingly, a combination of CP and ZMAb in EVD patients resulted in a synergistic effect [242].

ZMapp, an antibody cocktail, is product of collaboration between US Mapp Biopharmaceutical, Inc. and Defyrus, Inc., This cocktail consists of the best mAb (13C6, 2G4 and 4G7) amongst the individual monoclonal antibodies which are present in Mab-003 and ZMAb [227,232]. When used to treat EBOV infected NHP, a 100% efficacy was demonstrated, even when used 120 h post infection [227] (Table 2). Compassionate use along with other treatments resulted in saving life of two US healthcare workers [243]. Additionally, during 2014–2016 outbreak, 25 patients received ZMAb or ZMapp as compassionate drugs along with other treatments where 88% patients survived [244]. ZMapp completed phase I clinical trial (CT: NCT02363322) in 2019 and is currently a part of phase II/III clinical trial (CT: NCT03719586).

MIL77E is a mAb cocktail consisting of two Chinese hamster ovary-optimized ZMapp antibodies (13C6 and 2G4; optimized versions named MIL773 and MIL771 respectively) [228]. It demonstrated a 100% protection in infected NHP when administered 72 h post infection [228] (Table 2).

Autophagy seems to play a significant role in EBOV infection [23] and can be one of therapeutic targets. Several microRNAs identified in the black flying fox (natural reservoir for EBOV) were suggested to target autophagy controlling genes [245]. Autophagy induction by type-I IFN signaling pathways was suggested as the mechanism behind protection of NHP against lethal EBOV infection upon administration of eVLP consisting viral proteins [246]. A recent study advocated a critical role of autophagy associated proteins such as microtubule-associated protein 1A/B light chain 3B (LC3B) in EBOV uptake [247]. Further studies are needed to contemplate the development of therapeutic measures targeting such proteins.

Overall, various therapeutic interventions present their specific advantages (reviewed in [244]). Griffithsin exhibits desirable host toxicity profile and has a synergistic effect with other antiviral drugs [215]. Favipiravir is readily available and can be orally administered [244]. TKM-100802 presents with low manufacturing costs and requirements [244]. CP is an economic intervention which is safe, outbreak strain specific and can restore the blood volume [230,231]. ZMapp offers desirable safety and immunogenicity while MIL-77E presents a more cost-effective alternate to ZMapp [232].

Undesirable cytokine induction may present challenges during TKM-100802 treatment [244]. Efficacy of Favipiravir seems to rely on viral titers [244]. Compound 7, NSC62914, LJ001, Tolcapone, 3-deazaneplanocin A and FGI-103, FGI-104 and FGI-106 need to be tested in NHP models before a possible advancement into human trials. The major limitations of various monoclonal antibody cocktail treatment approaches are (a) high production costs (b) extensive manufacturing time (c) requirement for protocol optimization and (d) possible absence of pan-ebolavirus protection (reviewed in [232]).

## 6. Open Questions

EVD is an emerging zoonosis mainly caused by EBOV. The virus can use various routes to enter the body, while the main portal of entry remains largely unknown [17]. Studies have suggested that the virus can enter via the skin, even without cuts or abrasions [248]. Tissue-resident DCs, patrolling monocytes (CD16^+^ monocytes) and inflammatory DCs were shown to contribute to virus dissemination [17,249]; however, the exact cell types, especially DC subsets, responsible for the early virus spread remain unknown [17]. Additionally, the role of DC-SIGN expressing DC in virus spread and the role of langerin expressing DC cells (resistant to EBOV infection) in adaptive immunity requires further investigation [17,250]. Identification of the target cell and initial steps of virus replication are essential for development of an effective therapeutics and vaccines.

EBOV infection significantly dysregulates the immune system where it could induce an immune suppression (apoptosis, immune evasion mechanisms, lymphopenia etc.) [67,69] as well as activation (T cell activation, inflammation) [58,62,66]. The leukocyte apoptosis mechanism [23] as well as role of proliferated T cell populations during the EBOV infection [17] are yet to be clarified. Additionally, it remains unknown whether a polyfunctional T cell response could result in protective immunity or worsens the disease manifestation [28]. Additionally, virus persistence in immune privileged sites even after a complete recovery has been reported to cause reoccurrence of EVD [251], by unknown mechanism [17].

Activation of TLR and IFN-I pathways can induce the autophagy protecting against infection [246]. Additionally, simulation of two-pore channels (TPCs) by utilizing nicotinic acid adenine dinucleotide phosphate (NADDP) induces autophagy and helps the viral spread [252]. Hence, role of autophagy, involved proteins such as Rab 7 [252] and autophagy antagonists such as tetrandrine, rapamycin etc., in host response and as plausible therapeutic targets needs to be analyzed [23].

Interestingly, studies suggest that rodents, some reptiles and fruit bats are naturally immune to EBOV infection [17,253]. One of the reasons for their EBOV resistance is the NPC1 polymorphism, a protein essential for the virus entry [254]. Another plausible reason is the identification of several microRNA involved in upregulation of damage response and autophagy genes in response to disease inducing activities of the virus in *Pteropus alecto* [245]. Still, the reasons behind the high susceptibility of humans to EBOV as compared to other species require further investigation [17].

## 7. Conclusions

In conclusion, studies have demonstrated a central role of innate and adaptive immune responses during EVD. Survivor cases are marked by an early but controlled cytokine production, early T-cell activation, a recovery from a bystander T-cell apoptosis, intact CD3 T-cell population and increased serum levels of RANTES, CD40L and CD28 transcripts along with development of anti-EBOV antibodies. It appears that significant and polyclonal T-cell as well as humoral immune responses are indispensable for survival. Some viral proteins help to evade the host immune responses by displaying antigenic subversion, steric occlusion, anti-tetherin activity, shielding viral genome for RIG-I and MDA-5 and/or restricting type I and II IFN response via interference with JAK-STAT and MAPK pathways. Therapeutic and vaccine development approaches targeting different viral proteins and/or pathogenic mechanisms have demonstrated fruitful results in animal studies as well as in clinical trials. Still there is no single drug has been identified to have a cross-reactive, globally protective and easily accessible measure against EBOV.

## Figures and Tables

**Figure 1 pathogens-09-00850-f001:**
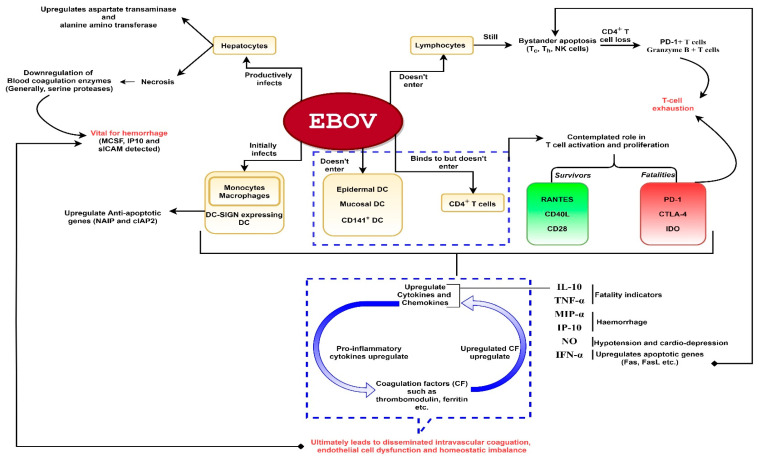
A schematic representation of Ebola virus (EBOV) pathogenesis. The virus is capable of productively infecting monocytes, macrophages, dendritic cells (DC) (except epidermal, mucosal and CD141^+^ DC) and hepatocytes. The virus does not enter lymphocytes but can show interaction with CD4^+^ T-cells. Despite non-entry into lymphocytes, bystander lymphocyte apoptosis is observed during the course of infection which could lead to T-cell exhaustion. Infection of hepatocytes could result in downregulation of blood coagulation enzymes which could lead to hemorrhage. Severe infection leads to hyperproduction of proinflammatory cytokines. These cytokines activate coagulation factors such as thrombomodulin, ferritin etc. The released coagulation factors in turn upregulate proinflammatory cytokines, as depicted. Hence, a deadly chain reaction ensues upon filoviral infection which might culminate into shock, vascular damage (disseminated intravascular coagulation which might lead to hemorrhage especially rashes, gastrointestinal and conjunctival hemorrhage in the later stages) and homeostatic imbalance.

**Figure 2 pathogens-09-00850-f002:**
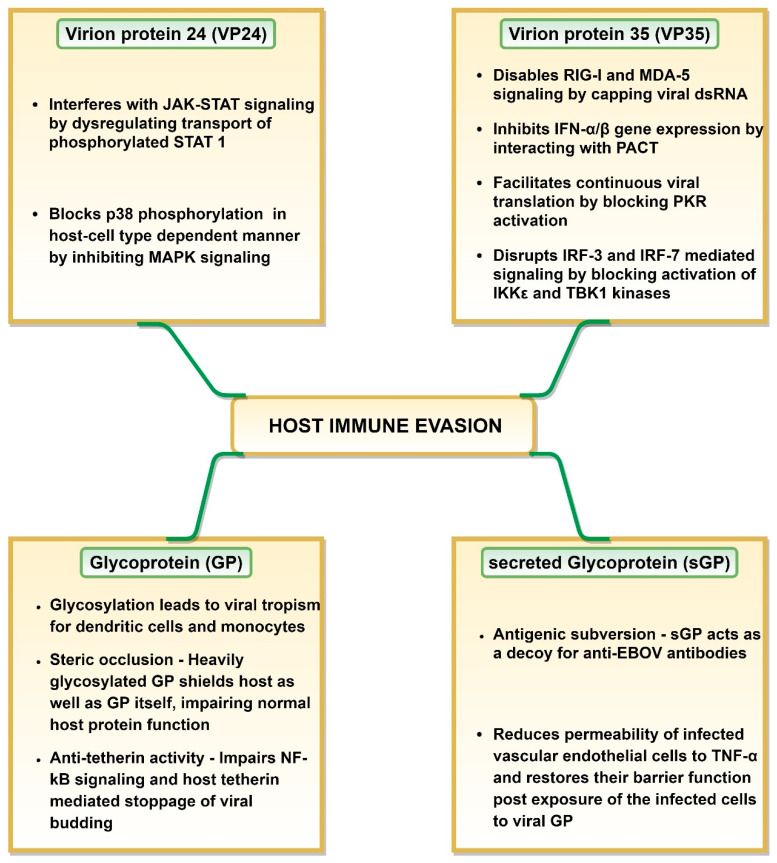
Mechanisms of host immune evasion employed by Ebola virus.

**Table 1 pathogens-09-00850-t001:** Vaccines development against Ebola virus under consideration and their clinical trial phase.

Vaccine	Responsible Party	Location	Clinical Trial Phase	References
INO-4212	Inovio Pharmaceuticals	United states	I	[147]
rChAd3-EBOS	NIAID	Uganda	I	[148]
DNA plasmid vaccine (VRC-EBODNA012-00-VP)	NIAID	United states	I	[149,150]
HPIV3-EBOVZ GP	NIAID	United states	I	[151]
HPIV3/ΔHNF/EbovZ GP	NIAID	United states	I	[152]
ChAd3-EBO-Z + Ad26.ZEBOV	University of Oxford	United Kingdom	I	[153]
rVSVN4CT1-EBOVGP1	Auro vaccines LLC	United states	I	[146]
rChAd3-EBOV + MVA BN-Filo	University of Oxford	Senegal	I	[154]
rAd5-EBOV	Sierra Leone-China Friendship Hospital	Sierra Leone	II	[155]
rChAd3-ZEBOV	GlaxoSmithKline	Mali and Senegal	II	[156,157]
rAd26-EBOV + MVA-BN-Filo + rVSVΔG-ZEBOV-GP	NIAID	Guinea, Liberia, Mali and Sierra Leone	II	[158]
VSVG-ZEBOV + ChAd3-EBO Z	NIAID	Liberia	II	[159]
MVA-BN Filo + rAd26-EBOV	London School of Hygiene and Tropical Medicine	DRC	III	[160]
GamEvac-combi vaccine	Gamaleya Research Institute of Epidemiology and Microbiology	Guinea and Russian Federation	IV	[161,162]

**Table 2 pathogens-09-00850-t002:** Therapeutic drug development measures against Ebolaviruses.

Drug Name	Nature	Responsible Party, Location *	Target	Clinical Trial Phase/Efficacy	Reference
Mannose-binding lectin (MBL)	Carbohydrate-binding protein (or lectin)	-	GP	40% in mice	[216]
Griffithsin (GRFT)	Carbohydrate-binding protein (or lectin)	-	Glycan structures	-	[215]
BCX4430	Adenosine analog	BioCryst Pharmaceuticals, United States	RNA polymerase	I	[197,217]
T-705 (Favipiravir)	Pyrazinecarboxamide derivative	Institut National de la Santé Et de la Recherche Médicale, Guinea	RNA polymerase	II	[218]
Aptamers	Oligonucleotide	-	VP35	-	[219]
AVI-6002 (PMO)	Phosphorodiamidate Morpholino Oligomer	Sarepta Therapeutics, Inc., United States	VP24 and VP35	I/60% in NHP	[194,220]
TKM-100802	Small interfering RNA	Arbutus Biopharma Corporation, United States	RNA polymerase and VP35	Trials terminated/100% in NHP	[191,221]
Small molecule inhibitor of VP40	-	-	VP40	-	[222]
GS-5734	Adenosine triphosphate analog	NIAID, Guinea and Liberia	RNA synthesis	II/100% in NHP when administered 72 h post infection	[202,223]
3-Deazaneplanocin A	Carbocyclic nucleoside	-	RNA synthesis	100% in BALB/c mice when administered 48 h post infection	[212]
Mab114	Monoclonal antibody	NIAID, United States	GP	Safe and well tolerated in phase I trial	[224]
MB-003	Monoclonal antibody cocktail	-	GP	120 h delayed intervention protected 40% NHP	[225]
U18666A	Cationic sterol	-	ebolavirus-NPC1 interaction	Almost 100% efficacy in vitro	[38]
ZMAb	Monoclonal antibody cocktail	-	GP	100% in NHP given three doses in six days starting 24 h after infection	[226]
ZMapp	Monoclonal antibody cocktail	NIAID, United States, Guinea, Liberia and Sierra Leone	GP	Completed phase I/100% in NHP when administered five days post-infection	[227]
MIL77E	Monoclonal antibody cocktail	-	GP	100% in NHP when give 72 h post infection	[228]

* Responsible party is the collaborator responsible for clinical trial. Location specifies the area where clinical trial (if any) was conducted. In case no clinical trial has been reported as yet, no responsible party or location has been mentioned.

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
