# Peer review of "Immunological Perspective for Ebola Virus Infection and Various Treatment Measures Taken to Fight the Disease"

_pathogens, 2020, doi:10.3390/pathogens9100850_

Round 1

Reviewer 1 Report

This review of Ebola virus (EBOV) pathogenesis is an excellent addition to Pathogens. Overall, this is an extremely well-written review, with an impressive in-depth EBOV immunology and vaccination/therapeutics section. The tables and figures are thorough, informative, comprehensive and extremely well-articulated.

A concern with the review is the discordance between the immunology sections, questioning the importance of humoral immunity and some aspects of T cell immunity in EBOV disease progression; and the subsequent sections on vaccination, where the authors use these same immunology correlates (outlined as not efficacious in the immunology sections) as demonstrating that vaccine or therapies may be advantageous.

The T cell immunology section could benefit from some more organization in the interest of clarity for the general readership. In this section, the authors summarize compelling data illustrating the importance of T cells in EBOV pathogenesis and protection, while making more subtle points regarding T cells not being a correlate/biomarker for EBOV disease progression. The distinction between these two points is quite subtle for a general Pathogens audience. Delineating these two points for clarity is important for the readership. Organizing this section to summarize data regarding T cells in protection from and limiting of EBOV infection, followed by a paragraph summarizing the lack of T cell response as a correlate/biomarker for EBV progression. Otherwise, introducing this section to indicate that T cells are not a biomarker of EBOV disease progression, then going into how important T cells are in EBOV pathogenesis is confusing.

The authors tackle in-depth analyses and subtleties of immune responses to EBOV in their review, which is highly admirable and productive. However, this should then be related to the vaccination and therapy studies reviewed, which may not have, at the time of their reporting, discussed the subtleties of EBOV disease progression and immunopathogenesis. For this reason, I suggest that the authors weigh-in more about the pros and cons of the therapies and vaccination strategies that they review rather than report on them without editorial.

Other major comments:

Lines 314-315 While it is stated that an antibody response is elicited by EBOV infections, it is previously discussed in the manuscript that GP is glycosylated and evades humoral immunity. Even if antibodies are elicited, they very well might not be protective.  This statement is in contrast to earlier statements laid out in the manuscript regarding the potential limitations of antibody responses in EBOV, e.g., will the virus be a target of neutralizing antibodies if GP is glycosylated in the manner described in the earlier sections of the manuscript on humoral immunity?

Lines 316-317 Cytokine overproduction is a problem in EBOV pathogenesis as described earlier in the manuscript. Thus, the statement that cytokine production is induced by the vaccine is not so informative, as this can be an aspect of EBOV pathogenesis vs. protection. Are the levels similar to those induced by natural infection? If so, this could be a concern. Again, earlier in the text, the detrimental effects of cytokine production after EBOV infection are highlighted (and come across as inconsistent with the statements in this section on vaccination). If the authors are trying to say that there is some immune response being induced by the vaccine, simply say this, but add the caveats as to the aspects of EBOV immunopathogenesis that were discussed earlier in the manuscript.

Line 325   The vaccine-elicted IFN-gamma production in mononuclear cells does not necessarily equate to a beneficial response. For example, these mononuclear cells are short-lived and are not part of immunologic memory. How these cells impact the primary T cell immune response is important, but nothing is mentioned about what T cell populations are induced by this vaccine. Since mononuclear cells are a dominant permissive cell type for EBOV infection and are short-lived (and not associated with immunological memory), the importance of this statement needs to be qualified. IgG levels are reasonable to report. T cell responses are left out of this section, therefore, the mononuclear cell IFN-gamma production and why this is important should be noted.

The third paragraph in the “Open Questions” section (no line numbers) on autophagy comes out of nowhere. It is not mentioned in the manuscript until this section. This reviewer recommends for autophagy to be discussed earlier in the manuscript, and then summarized in the “Open Questions” section. Perhaps this paragraph on autophagy can be included in the therapeutics section or evasion of immunity section and then one or two of the outstanding questions listed in this “Open Questions” section.

In summary, this is an excellent and extremely well-written review, covering important and somewhat complex aspects of what we know about EBOV vaccines, immunity, cell biology and immune evasion, and therapies. I recommend its acceptance to Pathogens.  Minor comments are included below.

Minor comments:

General Comment: I would specify when the cited paper is a review (i.e., References 17, 18). I would state “reviewed in [17, 18]”

Lines 100-101.  This line is a little confusing. “The fusion process can be blocked by the cysteine protease determents and acidic pH of the environment.[41]” Do you mean by increasing pH?  Clarifying this statement by elaborating on what is known about pH and viral entry would be useful.

Lines 138-145    This reviewer thinks commenting on increased lymphocyte apoptosis and elaborating about this in more detail would be of interest to readership.

The T cell section is a little confusing. Perhaps the authors can separate protection and lack of biomarkers (see Comments to Authors).

Line 160  “adoptive”, not adaptive transfer

Line 200 “uninfected” instead of “non infected”

Line 203 “Models” or “a” mouse model

Table I. The following, below, are unclear:

“Enshrouding of receptor- binding and fusion domains” is a little unclear. This could be more specific, i.e., soluble GP acts as a decoy for antibodies.

“The effect of antibodies is reduced drastically and probably completely upon exposure of vital functional domains only transiently upon entry.” Define vital functional domains.

Lines 331-2  “Moved toward to be tested” is awkward. “is moving forward to being tested” instead?

Table 3   U18666A – Percent efficacy is missing in Table 3.

Line 349 “In a study, only one third of the total EBOV exposed Rhesus macaques survived upon administration of rNAPc2.” This is unclear since it is not stated, in the absence of rNAPc2, how many macaques died. Assuming it is 100%, then stating “only partial and limited protection is elicited by rANPc2” would be a clearer statement.

Page 3 of 29 has no line numberings and therefore difficult for this reviewer to reference the subsequent lines. Page numbers are listed in reverse order and it is difficult to reference page numbers.

The third paragraph in the “Open Questions” section (no line numbers) on autophagy comes out of nowhere. This reviewer recommends for autophagy to be discussed earlier in the manuscript, and then summarized in the “Open Questions” section. (see Comments to Authors)

Line 5 under Conclusions (no line numbers). The authors use the term “well-balanced”. The meaning of well-balanced is unclear. It would be of benefit to be more specific about this term, i.e., highlight which T cell characteristics are aspects of what is considered “well-balanced” by the authors.

Line 5 under Conclusions (no line numbers)   “aa”    This appears to be a typo.

Author Response

We appreciate the reviewers for their valuable comments in improving the quality of manuscript. Response to specific comments raised by the reviewers is given below, which are incorporated in the revised manuscript as yellow highlight.

Reviewer 1

Comments

Query 1 a): The T cell immunology section could benefit from some more organization in the interest of clarity for the general readership. In this section, the authors summarize compelling data illustrating the importance of T cells in EBOV pathogenesis and protection, while making more subtle points regarding T cells not being a correlate/biomarker for EBOV disease progression. The distinction between these two points is quite subtle for a general Pathogens audience. Delineating these two points for clarity is important for the readership. Organizing this section to summarize data regarding T cells in protection from and limiting of EBOV infection, followed by a paragraph summarizing the lack of T cell response as a correlate/biomarker for EBV progression.

Query 1 b): Lines 138-145 This reviewer thinks commenting on increased lymphocyte apoptosis and elaborating about this in more detail would be of interest to readership. The T cell section is a little confusing. Perhaps the authors can separate protection and lack of biomarkers (see Comments to Authors).

Response 1 a) & b):  We agree with reviewer remark. Now, section 3.2. is more organized and T cell response sub-section is followed by lymphocyte apoptosis sub-section. Some data on lymphocyte apoptosis has been added. A paragraph, earlier mentioned in T-cell response sub-section, is now mentioned appropriately in the apoptosis section indicating the significant role of T cells so as to present the data in a simple manner keeping in mind general readership.

3.2. section is as follows:-

Line 131 – “3.2. Cell mediated immune response

3.2.1. T-cell response

Multiparametric flow cytometry analysis has revealed a robust activation of Tc cells followed by a substantial proliferation in fatal as well as survivor cases. [28,56,58]. An average of 45% CD8+ T cells, consisting of HLA-DR+/CD38+, Ki-67+/granzyme B+ and Ki-67+/PD-1+ subsets, were found to express activation markers HLA-DR, Ki-67 and CD38 [56]. NHP studies focused on EBOV GP presented contrasting results regarding the indispensability of CD8+ T cells for EBOV infection survival [59,60]. In 2018, Sakabe and coworkers demonstrated that memory CD8 T cells secrete interferon gamma (IFN-γ) and tumor necrosis factor alpha (TNF-α) in nearly 80% survivor subjects, especially upon activation with proteins other than GP viz., EBOV nucleoprotein, virion protein 24 (VP24) or VP40 [61]. Analysis of samples collected during the West African epidemic (2014-2016) by Speranza and co-workers revealed the abundance of T-cell immunity transcripts (RANTES, CD40L, CD28 etc.) in survivors and of T cell homeostasis drivers (PD-1 and Indoleamine 2,3-dioxygenase) in fatal cases, thus, confirming the notion of a robust and sustained T-cell response mounted in survivors [62]. Moreover, survivors show a characteristic chemokine (C-C motif) ligand 5 (CCL5/ RANTES) expression, further, supporting the role of T cells in viral clearance. In contrast, the fatal cases are devoid of T cell viral clearance and present a clustering of T cells in gut and respiratory mucosa [62].

Immunophenotyping analysis of fatal and survivor blood samples indicated activation of CD8+ and CD4+ T cells. Still, the magnitude and diversity of the immune response induced in the survivors were more robust as compared to fatal cases. Proliferation of T cells and yet their failure to effectively protect against EVD in fatal cases may be attributed to either a state of T-cell exhaustion [58,63,64] or little and delayed proliferation in some cases owing to exaggerated viral count and uncontrolled viral replication [28,56]. Also, activation of Ki-67+/PD-1+ CD8+ T cell subset seems responsible for a weaker adaptive immune response via PD-1 inhibitory pathway [56]. Indeed, oligoclonal response and greater expression of CTLA-4 and PD-1 in CD8+ and CD4+ T cells was found in fatal cases, which could explain the high virus titer as well as T-cell exhaustion [58,62,63,65,66]. Whether the high viral titers and inflammation could cause a greater CTLA-4 and PD-1 expression on T cells in fatal cases remains unclear [58].

3.2.2. Lymphocyte apoptosis

A massive reduction of lymphocyte (CD8+, CD4+ and NK cells) counts was found in the initial as well as the end stages of fatal EVD [67]. Experimentally infected NHP models displayed a reduction in peripheral NK cell count which may be attributed to apoptosis [68]. CD8+ and CD4+ cells appear to be the most affected as their counts in fatal cases were found reduced to an approximately one fourth their number in survivors [23]. Expression analysis of CD95 for CD8+ and CD4+ T cells as well as PD-1 for CD4+ T cells in EVD patients suggested the role of apoptotic pathways in massive lymphocyte loss [69]. Lymphocyte apoptosis could be attributed to a) deregulation of DC/T interaction ,i.e., lack of co-stimulatory rescue signals by malfunctioning DCs or b) upregulation of apoptotic genes such as Fas, Fas Ligand (FasL) and tumor necrosis factor (TNF)-related apoptosis-inducing ligand (TRAIL) in infected leukocytes or c) Direct lysis by EBOV GP [23,67,70]. Indeed, excess TNF-α secretion is thought to contribute to lymphocyte apoptosis in NHP models [71].

However, bystander T-cell apoptosis doesn’t seem to be a definite EVD characteristic and as such, does not seem to indicate fatality. This is supported by a report that EBOV caused death in apoptotic gene knockout mice despite reduced T cell apoptosis [72]. Also, T cells produced in experimentally EBOV infected mice were found to protect naive mice upon adoptive transfer [73] while complete depletion of T cells in experimentally infected NHP led to an increased fatality rate [59] suggesting a significant role of T cells in host survival. Similar data was demonstrated using mice deficient in cytotoxic T cells (Tc cells), where mice were dying upon experimental EBOV infection [74].

Overall, various reports suggest a critical, though highly varied role of T cells/cell mediated immune response upon EBOV infection in both, fatal and survivor cases and therefore, the actual lymphocyte apoptosis mechanism, T cell immunity dynamics and behavior during EVD is vital to understanding EBOV pathology.”

Query 2: The authors tackle in-depth analyses and subtleties of immune responses to EBOV in their review, which is highly admirable and productive. However, this should then be related to the vaccination and therapy studies reviewed, which may not have, at the time of their reporting, discussed the subtleties of EBOV disease progression and immunopathogenesis. For this reason, I suggest that the authors weigh-in more about the pros and cons of the therapies and vaccination strategies that they review rather than report on them without editorial.

Response: We have now discussed the advantages and limitations of various preventive measures in the manuscript. The additions are as follows:-

Line 395 – “To summarize, the vaccine approaches discussed so far present specific advantages. Non-replicating vectors such as Ad5 and EBOV∆VP30 are considerably safer than replication-competent vectors (reviewed in [182]). rVSV vector based vaccines are highly immunogenic and seem to be devoid of pre-existing immunity concerns [145,151]. A combination of MVA-BN Filo and rAd26-EBOV seems capable of generating both, cell mediated and humoral immune response [155]. H2O2 treated EBOV∆VP30 vaccine candidate addresses the earlier safety concerns and has exhibited high experimental efficacy [182] while recombinant rabies virus-based vector offers an opportunity to fight against two diseases (Rabies and EBOV) endemic to same geographical regions [174]. Peptide-based vaccines offer a safer, economical and faster approach to counter the rapidly increasing threat of endemic pathogens [176].

These vaccine development approaches also suffer from different limitations (reviewed in [183]). Production difficulties and high cost are the major limitations of using the VLP platform [182]. DNA vaccines require regular booster dosages (reviewed in [184]). Pre-existing immunity against adenovirus based vectors and consequently, low efficacy, presents a major concern [185]. Low immunogenicity and in vivo peptide instability are major challenges while designing peptide-based vaccines [176].”

Line 446 – “However, despite experimental success, low efficacy in human trials has been observed [205].”

Line 534 – “Overall, various therapeutic interventions present their specific advantages (reviewed in [244]). Griffithsin exhibits desirable host toxicity profile and has a synergistic effect with other antiviral drugs [215]. Favipiravir is readily available and can be orally administered [244]. TKM-100802 presents with low manufacturing costs and requirements [244]. CP is an economic intervention which is safe, outbreak strain specific and can restore the blood volume [230,231]. ZMapp offers desirable safety and immunogenicity while MIL-77E presents a more cost-effective alternate to ZMapp [232].

Undesirable cytokine induction may present challenges during TKM-100802 treatment [244]. Efficacy of Favipiravir seems to rely on viral titers [244]. Compound 7, NSC62914, LJ001, Tolcapone, 3-deazaneplanocin A and FGI-103, FGI-104 and FGI-106 need to be tested in NHP models before a possible advancement into human trials. The major limitations of various monoclonal antibody cocktail treatment approaches are a) high production costs b) extensive manufacturing time c) requirement for protocol optimization and d) possible absence of pan-ebolavirus protection (reviewed in [232]).”

Query 3: Lines 314-315 While it is stated that an antibody response is elicited by EBOV infections, it is previously discussed in the manuscript that GP is glycosylated and evades humoral immunity. Even if antibodies are elicited, they very well might not be protective. This statement is in contrast to earlier statements laid out in the manuscript regarding the potential limitations of antibody responses in EBOV, e.g., will the virus be a target of neutralizing antibodies if GP is glycosylated in the manner described in the earlier sections of the manuscript on humoral immunity?

Response: Even though the virus employs various immune evasion mechanisms, an immune response (innate as well as adaptive), albeit of varying degree, is generated in both, fatalities as well as survivors. Also, generation of both, cell mediated and humoral immune response is thought to be the best marker of survival. These statements have been mentioned in the manuscript. However, to further clarify the statements under consideration, the following modifications have been made:-

Line 348 – “Ervebo vaccine induces antibody response and CD8+ T-cell activation upon administration [152], though the efficacy of resultant antibodies in viral clearance needs further analysis.”

Query 4: Lines 316-317 Cytokine overproduction is a problem in EBOV pathogenesis as described earlier in the manuscript. Thus, the statement that cytokine production is induced by the vaccine is not so informative, as this can be an aspect of EBOV pathogenesis vs. protection. Are the levels similar to those induced by natural infection? If so, this could be a concern. Again, earlier in the text, the detrimental effects of cytokine production after EBOV infection are highlighted (and come across as inconsistent with the statements in this section on vaccination). If the authors are trying to say that there is some immune response being induced by the vaccine, simply say this, but add the caveats as to the aspects of EBOV immunopathogenesis that were discussed earlier in the manuscript.

Response: We agree with the reviewer. For clarity and better understanding of a general reader, the information on cytokine response generated has been replaced with information on T cells. The statement has been modified as follows:-

Line 350 – “Before FDA approval of Ervebo vaccine, Farooq et al., reported a role of circular follicular T helper cells upon injection of rVSV-ZEBOV-GP vaccine candidate in human subjects [154]. A correlation between the frequency of circulating CXCR5+ CD4+ T-cells and antibody titers was also recorded [154].”

Query 5: Line 325 The vaccine-elicted IFN-gamma production in mononuclear cells does not necessarily equate to a beneficial response. For example, these mononuclear cells are short-lived and are not part of immunologic memory. How these cells impact the primary T cell immune response is important, but nothing is mentioned about what T cell populations are induced by this vaccine. Since mononuclear cells are a dominant permissive cell type for EBOV infection and are short-lived (and not associated with immunological memory), the importance of this statement needs to be qualified. IgG levels are reasonable to report. T cell responses are left out of this section, therefore, the mononuclear cell IFN-gamma production and why this is important should be noted.

Response: As sufficient data on T cell response or the significance of IFN-γ producing mononuclear cells could not be found in the study conducted on NHP model by Marzi et al., 2015, we have now reviewed their study as follows:-

Line 367 – “Wild type as well as H2O2 treated EBOV∆VP30 vaccine conferred complete protection via producing high antibody titers directed against various viral proteins [136]. Production of IFN-γ producing mononuclear cells was also suggested though their significance and impact was not reported [136].”

Query 6 a): The third paragraph in the “Open Questions” section (no line numbers) on autophagy comes out of nowhere. It is not mentioned in the manuscript until this section. This reviewer recommends for autophagy to be discussed earlier in the manuscript, and then summarized in the “Open Questions” section. Perhaps this paragraph on autophagy can be included in the therapeutics section or evasion of immunity section and then one or two of the outstanding questions listed in this “Open Questions” section.

Query 6 b): The third paragraph in the “Open Questions” section (no line numbers) on autophagy comes out of nowhere. This reviewer recommends for autophagy to be discussed earlier in the manuscript, and then summarized in the “Open Questions” section. (see Comments to Authors).

Response 6 a) & b): As correctly pointed out by the reviewer, we have now added a paragraph on autophagy in the therapeutics section as follows:-

Line 526 – “Autophagy seems to play a significant role in EBOV infection [23] and can be one of therapeutic targets. Several microRNAs identified in the black flying fox (natural reservoir for EBOV) were suggested to target autophagy controlling genes [245]. Autophagy induction by type-I IFN signaling pathways was suggested as the mechanism behind protection of NHP against lethal EBOV infection upon administration of eVLP consisting viral proteins [246]. A recent study advocated a critical role of autophagy associated proteins such as microtubule-associated protein 1A/B light chain 3B (LC3B) in EBOV uptake [247]. Further studies are needed to contemplate the development of therapeutic measures targeting such proteins.”

Query 7: I would specify when the cited paper is a review (i.e., References 17, 18). I would state “reviewed in [17,18]”

Response: The changes have been made throughout the manuscript. Some examples are mentioned below:-

Line 52 - (reviewed in [17,18])

Line 223 - (reviewed in [23])

Line 458 - (reviewed in [214])

Query 8: Lines 100-101. This line is a little confusing. “The fusion process can be blocked by the cysteine protease determents and acidic pH of the environment.[41]” Do you mean by increasing pH? Clarifying this statement by elaborating on what is known about pH and viral entry would be useful.

Response: The statements have been clarified as follows:-

Line 96 – “A low pH dependent endosomal function is required by cathepsin digested GP1 subunit for fusion [41]. Therefore, fusion process can be blocked by the cysteine protease determents and increasing the pH of the environment [41].”

Query 9 a): Line 160 “adoptive”, not adaptive transfer

Query 9 b): Line 200 “uninfected” instead of “non infected”

Query 9 c): Line 203 “Models” or “a” mouse model

Query 9 d): Lines 331-2 “Moved toward to be tested” is awkward. “is moving forward to being tested” instead?

Query 9 e): Page 3 of 29 has no line numberings and therefore difficult for this reviewer to reference the subsequent lines. Page numbers are listed in reverse order and it is difficult to reference page numbers.

Query 9 f): Line 5 under Conclusions (no line numbers) “aa” This appears to be a typo.

Response 9 a) - f): Changes have been made as rightly suggested by the reviewer. The formatting, typos and grammatical errors have been removed.

Query 10: Table I. The following, below, are unclear:

“Enshrouding of receptor- binding and fusion domains” is a little unclear. This could be more specific, i.e., soluble GP acts as a decoy for antibodies.

“The effect of antibodies is reduced drastically and probably completely upon exposure of vital functional domains only transiently upon entry.” Define vital functional domains.

Response: As per this query and suggestion by Reviewer 2, we have removed table 1 and instead, added a schematic representation of mechanisms followed by EBOV for host immune evasion. Therefore, this query has been addressed.

Query 11: Table 3 U18666A – Percent efficacy is missing in Table 3.

Response: As per this query and suggestion by Reviewer 2, efficacies have now been mentioned in table enlisting therapeutic drug development measures.

Query 12: Line 349 “In a study, only one third of the total EBOV exposed Rhesus macaques survived upon administration of rNAPc2.” This is unclear since it is not stated, in the absence of rNAPc2, how many macaques died. Assuming it is 100%, then stating “only partial and limited protection is elicited by rANPc2” would be a clearer statement.

Response: The suggestion has been implemented and the current statement is as follows:-

Line 417 – “In a study, only one third of the total EBOV exposed Rhesus macaques survived upon administration of rNAPc2, indicating only partial and limited protection elicited by rNAPc2 [188].”

Query 13: Line 5 under Conclusions (no line numbers). The authors use the term “well-balanced”. The meaning of well-balanced is unclear. It would be of benefit to be more specific about this term, i.e., highlight which T cell characteristics are aspects of what is considered “well-balanced” by the authors.

Response: By well-balanced, we wanted to implicate neither excessive nor scant. To clarify the statement, we have now presented it as follows:-

Line 581 – “It appears that significant and polyclonal T-cell as well as humoral immune responses are indispensable for survival.”

Reviewer 2 Report

The article by Sahil Jain et al. is a good reveiw on Ebola Virus Infection and therapeutic management of the disease.

The article deserve some revision before possible acceptance in Pathogens.

Major :

The figure 1 could be completed adding information about the blood coagulation enzayme, the implicated cytokines, the coagulation factors... The EBOV has to be indicated in the center of the figure to fully understand the crucial role of this infection. This adding will be need to summarize the understanding of the 3.3 paragraphes.

Part 4 could be summarized in an appropriate table but a scheme would be easier to understand, please modify.

Table 2 could be completed indicating the manufacturer (and country)

Table 3 must be completed adding : the nature of the Drug, the manufacturers and coordinate, the efficacy, and the limited. Moreover, the drug could be classified by efficacy rates to ease the understanding.

Minor : 

Since line 148, line numbering are inverted please correct

Line 154 : imbalan : imbalance ?

Line 158 : could the authors details the knockout mice model (gene? background?)

Line 295 : steric occlusion must be indicated between "

Line 315 : et al. must be indicated in italic/

Global :

Beware of the use of italic/non-italic or bold/non-bold police in the manuscript.

The pages 9-12 are in landscape mode. Even if the table need this presentation, please add the text in portrait mode.

Numbers below or equal to 12 must be written in full letters.

Author Response

Response to Reviewers comments

We appreciate the reviewers for their valuable comments in improving the quality of manuscript. Response to specific comments raised by the reviewers is given below, which are incorporated in the revised manuscript as yellow highlight.

Reviewer 2

Comments

Query 1: The figure 1 could be completed adding information about the blood coagulation enzayme, the implicated cytokines, the coagulation factors... The EBOV has to be indicated in the center of the figure to fully understand the crucial role of this infection. This adding will be need to summarize the understanding of the3.3 paragraphes.

Response: In the figure, EBOV is now a little to the center of the figure as much possible. Also, EBOV font and formatting was changed to highlight it more. The names of coagulation factors (thrombomodulin, ferritin), haemorrhage-associated molecules (MCSF, IP10 and sICAM) and blood coagulation enzymes (generally, serine proteases) are mentioned. New data added in the manuscript is as follows:-

Line 226 – “3.4. Interplay between cytokines and coagulation factors

As discussed, severe infection leads to hyperproduction of proinflammatory cytokines. These cytokines activate coagulation factors such as thrombomodulin, ferritin etc. [23]. A study reported upregulation of procoagulant protein tissue factor in endothelial cells and monocytes by TNF-α and IL-6 [89]. The released coagulation factors in turn upregulate proinflammatory cytokines. Studies have suggested that fibrin fragment E and thrombin induce IL-6 production in monocytes while thrombin induces IL-6 and IL-8 production in endothelial cells [18,90]. Hence, a deadly chain reaction ensues upon filoviral infection which culminates into shock, vascular damage and homeostatic imbalance.

3.5. Endothelial cell dysfunction and vascular damage

Endothelial cells seem to be directly infected during terminal stages of EBOV infection due to over expression of proinflammatory cytokines but do not seem to exhibit any structural damage [18,28]. EBOV GP, supported by TNF-α, is thought to play a pivotal role in endothelial cell dysfunction, consequently, leading to anoikis and hemorrhage [23,91-93]. A study during 1995 outbreak found antigens in endothelial cells in different body tissues [94]. A recent study supported endothelial dysfunction on basis of increased thrombomodulin, P-selectin and PE-CAM (all are markers of endothelial activation and dysfunction) in patients [28,50]. However, the precise timing and consequences of endothelial cell dysfunction are yet to be elucidated. An early study reported no antigen presence in endothelial cells [95]. Also, antigen multiplication in vascular endothelial cells in later stages (after appearance of hemorrhage) upon experimental EBOV infection of cynomolgus monkeys has been reported [18,26].

Disseminated intravascular coagulation (DIC) coupled with low platelet count as well as coagulation factor deficiency is known to occur during EBOV infection [96]. It has been debatably related to endothelial cell disorders, especially release of thrombomodulin by activated endothelial cells [18,28,84]. Various reports attribute the endothelial cell activation and consequent dysfunction as well as vascular damage to a) release of proinflammatory cytokines [75,77,97], especially TNF-α [98,99] and NO [87] or b) overexpression of cell surface tissue factor in monocytes and macrophages [26] or c) elevated levels of Von Willebrand factor (vWF), a protein which acts as mediator between platelets and endothelial cells [28].”

Query 2: Part 4 could be summarized in an appropriate table but a scheme would be easier to understand, please modify.

Response: We agree with the reviewer and thank him for the suggestion. The table has now been replaced with a schematic representation of various mechanisms employed by EBOV to evade host immune responses.

Query 3: Table 2 could be completed indicating the manufacturer (and country)

Response: New columns for responsible party and location have been added in the vaccine development against Ebola virus table as per suggestion.

Query 4: Table 3 must be completed adding : the nature of the Drug, the manufacturers and coordinate, the efficacy, and the limited. Moreover, the drug could be classified by efficacy rates to ease the understanding.

Response: Limitations and advantages are now discussed throughout the manuscript for various preventive measures. In the table for therapeutic interventions, we have now added a column for nature and responsible party+location. However, in case no human clinical trial has been reported as yet, responsible party and location have not been mentioned for such cases. Further, wherever efficacy is being reported, we have included. However in  in certain cases, the efficacy data could be found. For example, a) BCX4430 – No result and publication reported for phase I clinical trial https://clinicaltrials.gov/ct2/show/NCT02319772 b) Griffithsin – A broad spectrum antiviral with clinical trials for HIV c) Favipiravir – Clinical trial phase II has inconclusive data as per the publication (Sissoko et al., 2016). https://clinicaltrials.gov/ct2/show/NCT02329054 d) AVI-6002 – No result posted for phase trial I. https://clinicaltrials.gov/ct2/show/NCT01353027. e) TKM-Ebola – Trials were terminated. https://clinicaltrials.gov/ct2/show/NCT02041715 f) GS-5734 – Has completed phase II trials. No results available. https://clinicaltrials.gov/ct2/show/NCT02818582.

Query 5 a): Since line 148, line numbering are inverted please correct

Query 5 b): Line 154 : imbalan : imbalance ?

Query 5 c): Line 158 : could the authors details the knockout mice model(gene? background?)

Query 5 d): Line 295 : steric occlusion must be indicated between "

Query 5 e): Line 315 : et al. must be indicated in italic/

Query 5 f): Beware of the use of italic/non-italic or bold/non-bold police in the manuscript.

Query 5 g): The pages 9-12 are in landscape mode. Even if the table need this presentation, please add the text in portrait mode.

Query 5 h): Numbers below or equal to 12 must be written in full letters.

Response a) – h): The formatting, typos and grammatical errors have now been sorted.

For 5 c) - We have added the knockout gene as advised:-

Line 171 – “This is supported by a report that EBOV caused death in apoptotic gene knockout mice despite reduced T cell apoptosis [72].”

Reviewer 3 Report

This is a very nicely prepared review focusing on Ebola virus, highlighting the immunology and virology involved, as well as treatments available and the prospects for novel approaches.  It is well-referenced and well-written.  The authors have provided a useful resource for the field.

Author Response

Thanks to Reviewer.